# Core–Shell Fe_3_O_4_@C Nanoparticles for the Organic Dye Adsorption and Targeted Magneto-Mechanical Destruction of Ehrlich Ascites Carcinoma Cells

**DOI:** 10.3390/ma16010023

**Published:** 2022-12-20

**Authors:** Oxana S. Ivanova, Irina S. Edelman, Chun-Rong Lin, Evgeniy S. Svetlitsky, Alexey E. Sokolov, Kirill A. Lukyanenko, Alexander L. Sukhachev, Nikolay P. Shestakov, Ying-Zhen Chen, Aleksandr A. Spivakov

**Affiliations:** 1Kirensky Institute of Physics, Federal Research Center KSC Siberian Branch, Russian Academy of Sciences, Krasnoyarsk 660036, Russia; 2Institute of Engineering Physics and Radioelectronics, Siberian Federal University, Krasnoyarsk 660041, Russia; 3Department of Applied Physics, National Pingtung University, Pingtung City 90003, Taiwan; 4Laboratory of Biomolecular and Medical Technologies, Krasnoyarsk State Medical University Named after Prof. V.F. Voino-Yasenetsky, Krasnoyarsk 660022, Russia; 5Laboratory for Digital Controlled Drugs and Theranostics, Federal Research Center KSC Siberian Branch, Russian Academy of Sciences, Krasnoyarsk 660036, Russia

**Keywords:** magnetite nanoparticles, adsorption, organic dyes, aptamers, magnetically induced cell destruction

## Abstract

The morphology, structure, and magnetic properties of Fe_3_O_4_ and Fe_3_O_4_@C nanoparticles, as well their effectiveness for organic dye adsorption and targeted destruction of carcinoma cells, were studied. The nanoparticles exhibited a high magnetic saturation value (79.4 and 63.8 emu/g, correspondingly) to facilitate magnetic separation. It has been shown that surface properties play a key role in the adsorption process. Both types of organic dyes—cationic (Rhodomine C) and anionic (Congo Red and Eosine)—were well adsorbed by the Fe_3_O_4_ nanoparticles’ surface, and the adsorption process was described by the polymolecular adsorption model with a maximum adsorption capacity of 58, 22, and 14 mg/g for Congo Red, Eosine, and Rhodomine C, correspondingly. In this case, the kinetic data were described well by the pseudo-first-order model. Carbon-coated particles selectively adsorbed only cationic dyes, and the adsorption process for Methylene Blue was described by the Freundlich model, with a maximum adsorption capacity of 14 mg/g. For the case of Rhodomine C, the adsorption isotherm has a polymolecular character with a maximum adsorption capacity of 34 mg/g. To realize the targeted destruction of the carcinoma cells, the Fe_3_O_4_@C nanoparticles were functionalized with aptamers, and an experiment on the Ehrlich ascetic carcinoma cells’ destruction was carried out successively using a low-frequency alternating magnetic field. The number of cells destroyed as a result of their interaction with Fe_3_O_4_@C nanoparticles in an alternating magnetic field was 27%, compared with the number of naturally dead control cells of 6%.

## 1. Introduction

In recent decades, nanoscale materials have taken an increasingly important place, not only in engineering and technology but also in such areas of human activity as environmental protection or medical applications. Now, it is already impossible to solve, for example, the problems of water purification from a wide variety of pollutants without the use of nanoscale materials [1,2,3]. The adsorption of pollutants by the surface of nanoparticles is one of the main methods for purifying liquid substances, including water. Nanoparticles made of various materials and structures based on them showed high adsorption efficiency with respect to a large number of pollutants. Among other nanomaterials, tri-metallic layered double hydroxide (NiZnAl-LDH) nanosheets [4,5] and novel bimetallic (ZIF-8@ZIF-67) zeolitic imidazolate [6] were successively applied as adsorbents of rhodamine B and methyl orange. Natural montmorillonite modified with cetyl trimethyl ammonium bromide was used as an adsorbent for the removal of orange G anionic dye from aqueous solutions [7]. Chitosan modificated with 2,3-dihydroxy-benzaldehyde was successfully used for removing Rhodamine B dye from aqueous solution [8]. The authors of ref. [9] successively used the microorganism–graphene oxide (GO) composites for Uranium (VI) adsorption from aqueous solutions and revealed that Lysinibacillus-GO exhibited superior U (VI) removal ability. A Raphia–microorganism composite consisting of immobilized yeast cells Saccharomyces cerevisiae on Raphia farinifera fibers was studied in [10] as a sorbent for removing lead ions from aqueous solutions.

Considerable attention has recently been paid to magnetic nanoparticles (NPs). Being sometimes inferior to other types of nanoparticles in absorption capacity, magnetic NPs have the advantage that they are easily removed from a liquid medium using a magnetic field. In those situations, where the rapid removal of nanoparticles with adsorbed pollutants from water is critical, magnetic nanoparticles have no competitors [11,12,13]. An application of magnetic NPs as adsorbents is due to the ability of their surface to attach various compounds to themselves [1,14]. Having an indisputable number of advantages, the use of NPs faces certain difficulties, such as aggregation, surface contamination with substances from the atmosphere, adaptation of a laboratory experiment to large volumes, etc., and for the successful use of NPs on an industrial scale, all these problems must be solved. 

It is known that the ability of magnetic NPs, in particular magnetite, to adsorb substances of different nature is due to both the presence of hydroxyl groups on their surface and to coordinately unsaturated Fe^3+^ and Fe^2+^ ions. Depending on the nature and structure of the adsorbate molecules, they are easily retained on the active sites of the magnetite surface due to electrostatic, donor–acceptor, hydrophobic interactions, and/or the formation of hydrogen bonds. In the process of covalent immobilization, strong covalent bonds are formed between the modifier and the oxide surface. For these reasons, magnetite NPs are very convenient objects for covering them with various shells, creating core–shell structures, and functionalizing their surface [15,16,17,18]. Carbon is often used to synthesize core–shell magnetic structures. It is a classic material that has long been used as a sorbent, and an inert shell of various modifications of carbon not only protects the magnetic core from oxidation and degradation but also gives better adsorption properties to NPs’ structure. A number of works present data on the study of the adsorption of typical organic dyes of cationic and anionic types by magnetite–carbon nanostructures [19,20,21,22,23,24,25,26]. Note that in all the cases described, the adsorption capacity of magnetite–carbon nanostructured materials is less than that of activated carbon 190–260 mg/g [27], and they adsorb cationic dyes better. 

The wide spread in the adsorption capacity of magnetite nanoparticles and core–shell structure Fe_3_O_4_@C is due to individual surface properties depending critically on the synthesis technology. Usually, the nanoparticles’ adsorption is studied in relation to one type of pollutant, for example, to one type of dye. Study of the adsorption characteristics of the nanoparticles produced in one technological cycle with respect to various dyes, both cationic and anionic, undertaken in the first part of the work, seems to be a very useful task. Such a study could find the best ways to activate the NPs’ surface and select the proper candidates for specific applications. 

The second part of the work is devoted to studying the possibility of using the synthesized Fe_3_O_4_@C NPs with the modified surface in biomedical applications. Surface modification is widely used in biomedical applications, for example, surface-modified NPs are being studied for use as nanocarriers for targeted drug delivery [18,28,29], as sensors for detecting biomolecules [30], for antibacterial therapy [17,31], for the treatment of cancer and iron deficiency anemia [32], and for magneto-mechanical destruction of cancer cells in low-frequency alternating or rotating magnetic fields [33,34,35]. The latter application is being actively developed, and the selective attachment of nanoparticles to cells that need to be destroyed is facilitated by their functionalization with an aptamer [34,36,37,38]. Aptamers are short single-stranded DNA or RNA oligonucleotides and are often referred to as synthetic analogs of antibodies [39]. They are also widely used in cancer diagnostics and therapy [40]. 

Cultures of Ehrlich carcinoma cancer cells are the most accessible for experiments, and recently, they have become model objects for studying the destruction of cancer cells under various influences, including the action of a magnetic field [34,41]. So, in ref. [34], aptamer-conjugated superparamagnetic iron NPs coated with ferroarabinogalactan were used for targeted magneto-dynamic therapy of an Ehrlich carcinoma tumor in vivo. Ref. [41] presents magneto-dynamic therapy using the gold-coated magnetic NPs functionalized with DNA aptamers to selectively kill tumor cells in vivo. The search for various biocompatible shells that homogeneously cover the magnetic core and the development of methods for attaching specific aptamers to them in order to influence cancer cells is an urgent task of modern research.

The present work is related to both lines of research described above. We investigated the adsorption of two organic dyes belonging to the cationic group and two dyes of the anionic group by Fe_3_O_4_ and core–shell Fe_3_O_4_@C NPs, having previously studied their morphology, structure, and magnetic properties. Typical cationic (Methylene Blue (MB), rhodomine C (RhC)) and anionic (Congo Red (CR), and Eosin Y (EoY)) were chosen for investigation. We also functionalized Fe_3_O_4_@C NPs with aptamers and used them to carry out experiments on the magneto-mechanical disruption of Ehrlich ascite carcinoma cell cultures. 

## 2. Materials and Methods

### 2.1. Synthesis of Nanoparticles

Fe_3_O_4_ NPs were obtained as a result of the thermal decomposition of the iron–oleate complex. First, 120 mmol sodium oleate and 40 mmol iron (III) chloride hexahydrate were dissolved in a mixture of solvents (80 mL alcohol, 60 mL deionized water, and 140 mL n-hexane, C_6_H_14_) at 70 °C in air for 4 h and then cooled to room temperature. After cooling, the reaction product was divided into two layers. The top layer was an organic solution, the bottom layer was water. While mixing the two layers, 30 mL of deionized water was added to the mixture and the process was repeated three times, separating the sticky iron–oleate complex. Second, 40 mmol of iron–oleate complex and 20 mmole oleic acid (OA) were dissolved in 200 g of 1-Octadecene (ODE) at 320 °C under air for 3 h, and then cooled down to room temperature. After the mixture had cooled, 500 mL of ethanol was added to it. As a result, it again separated into two layers, the top layer was removed, and n-Hexane was added. A black material was precipitated and separated via applied magnetic field and washed several times with n-hexane. Finally, precipitate was dried at 30 °C for 6 h. The described process is schematically presented in Figure 1, top panel. 

To prepare carbon-coated NPs, 0.2 g of Fe_3_O_4_ NPs and glucose (1 g) were dispersed in distilled water (30 mL) by ultrasonication for 15 min. The mixture was stirred for 30 min and then sealed in a Teflon-lined stainless-steel autoclave (50 mL capacity). The autoclave was heated to and maintained at 200 °C for 12 h, and then was allowed to cool to room temperature. The black products were separated by externally applied magnetic field and washed several times with water and ethanol. Finally, they were dried at 60 °C for 6 h. This method of carbon coating was already used by us earlier to create a second layer of carbon on particles that already have a carbon shell [20]. The scheme of NPs’ carbon coating is shown in Figure 1, bottom panel. 

### 2.2. Functionalization of Fe_3_O_4_@C NPs by Aptamers

To use the obtained Fe_3_O_4_@C NPs in experiments with carcinoma cells, the carbon shell was chemically activated. First, the particles were treated in nitric acid to create carboxyl groups, similarly to the procedure used in [42] for the chemical oxidation of carbon-based shell magnetic NPs to improve colloidal stability. Then, they were activated similarly to the procedure used in [30] for the activation of the carboxyl group of carbon-coated nanotubes carboxylated by π–π stacking interaction, which were subsequently used for conjugation of aptamers. Next, the AS-42 aptamer was used, with the amino-primer AmPrcom5 complementary to the 5’-tail. This aptamer binds to Ehrlich ascitic carcinoma cells with high specificity and affinity [38]. The concentration of the aptamer (-nh2) in the buffer solution was 10 mM, and the incubation time was 30 min. NPs functionalized in this way with the aptamer were mixed with Ehrlich ascitic carcinoma cells for 30 min. To visualize the experiment, living cells were preliminarily tinted with trepan blue. The scheme of the functionalization of Fe_3_O_4_@C NPs by aptamers is shown in Figure 2. 

### 2.3. Dyes

Methylene Blue (C_16_H_18_ClN_3_S) is a thiazine cationic dye soluble in water and ethanol but insoluble in ether, benzene, and CHCl_3_. Its electronic and structural characteristics are close to those of the active groups of flavin enzymes. Therefore, experiments with MB can serve to model the functions of the active groups of enzymes.

Rhodamine C (C_28_H_31_ClN_2_O_3_) is the cationic fluoronic dye. Rhodamine dyes are generally toxic and soluble in water, methanol, and ethanol.

Congo Red (C_32_H_22_N_6_Na_2_O_6_S_2_) is the disodium salt of 4,4′-bis-(1-amino-4-sulfo-2-naphthylazo)biphenyl, a typical chemical anionic azo dye and indicator, slightly soluble in cold water, and easily soluble in hot water, ammonia, and ethyl alcohol. It is a chemical dye that binds to the β-amyloid protein that accumulates in neurons in Alzheimer’s disease.

Eosine Y (C_20_H_6_Br_4_O_5_Na_2_) is the xanthene anionic dye, which is soluble in water and is obtained by the action of bromine on fluorescein. It is dyed pink. It is believed that the Eosine molecules are able to absorb light in the visible region of the spectrum (spectral sensitization), which contributes to a significant improvement in the photovoltaic characteristics of solar cells coated with this dye.

### 2.4. Characteristic Methods

The synthesized NPs were examined with transmission electron microscope (TEM) JEOL JEM-1230 (JEOL Ltd., Tokyo, Japan) operated at an accelerating voltage of 80 kV (Precision Instruments Center of NPUST). The X-ray diffraction measurements were performed using a Bruker D8 Advance diffractometer (Bruker Corp., Billerica, MA, USA) with Cu Kα radiation, 40 kV, 25 mA, λ = 1.5418 Å.

Fourier transform infrared absorption (FT-IR) spectra were recorded with a VERTEX 70 (Bruker Optik GMBH, Leipzig, Germany) spectrometer in the spectral region of 400 ÷ 4000 cm^−1^ with the resolution 4 cm^−1^. The spectrometer was equipped with a Globar as the light source and a wideband KBr beam splitter and RT-DLaTG as the detector (Bruker Optik GMBH, Leipzig, Germany). For the measurements, round tablet samples of about 0.5 mm thick and of 13 mm in diameter containing NPs were prepared as follows: nanopowders in an amount lower than 0.001 g were thoroughly ground with 0.14 g of KBr; the mixtures were formed into tablets, which were subjected to cold pressing at 10,000 kg.

The magnetic properties were measured with the vibrating sample magnetometer (VSM 8600) Lakeshore (Lake Shore Cryotronics, Westerville, OH, USA).

Changes in the absorption spectra of the dye solutions were recorded with the UV/VIS circular dichroism spectrometer SKD-2MUF (OEP ISAN, Moscow, Russia). The quartz cells with optical path length of 5 mm were used.

For experiments with living cells, the aptamer-functionalized nanoparticles were incubated for 30 min with Ehrlich’s ascites carcinoma cells. The samples were placed in an alternating magnetic field H_max_ = 100 Oe at 50 Hz for 20 min. Photographs were taken with the Levenhuk M1400 PLUS (Levenhuk LLC, Fremont, CA, USA) microscope camera at 10× magnification 2 h after exposure to a magnetic field.

## 3. Results and Discussion

### 3.1. Morphology, Structure, and Magnetic Properties

The morphology of Fe_3_O_4_ NPs is presented in Figure 3a. They are well-dispersed NPs of about 15 nm in diameter with a narrow size distribution. XRD patterns (Figure 3b) showed that the magnetic core of both Fe_3_O_4_ and Fe_3_O_4_@C NPs has the spinel ferrite crystal structure, with the parameters of the most intense peaks corresponding to the Fe_3_O_4_ phase (PDF Card # 04-005-4319). The XRD peaks’ narrowing in the case of Fe_3_O_4_@C NPs comparing with Fe_3_O_4_ NPs indicates ordering of the outer layers of particles and/or improving the quality of crystallites in the volume of particles during the formation of the carbon shell when the Fe_3_O_4_ NPs are kept in a glucose solution for a long time.

The FT-IR spectra are shown in Figure 4. In the spectrum of pure Fe_3_O_4_ NPs, an intense band at 584 cm^−^^1^ was due to the Fe–O stretching vibrations (ν_1_-band) characteristic of magnetite [43,44,45,46]. This band was observed in the spectra of all samples. A splitted weaker band is seen for pure magnetite at 403 and 449 cm^−^^1^, corresponding to the ν_2_-band of the Fe–O linkage in magnetite [44]. Weak C=O bands are seen in the pure Fe_3_O_4_ NPs at 1625 and 1715 cm^−^^1,^ which is explained by the NPs’ synthesis with hexane. At the transition from magnetite NPs to core–shell Fe_3_O_4_@C NPs and then to core–shell NPs after treatment in nitric acid Fe_3_O_4_@C–COOH, these bands become stronger, confirming the correct identification. In the carbon-coated particles, C-O bands appeared at 1050 and 812 cm^−^^1^, the CH_2_-O-CH_2_ group was indicated by the appearance of a wide band in the region of 1150–1100 cm^−^^1^ [46]. In the nanoparticles treated in acid, the carboxyl group was clearly identified by the 1384 cm^−^^1^ band and CH_2_COOH by the 1475 cm^−^^1^ band. A slightly noticeable band at 798 cm^−^^1^ is characteristic of pendulum oscillations of the NH_3_ group and stretching vibrations –OH in the composition of the carboxyl group in the region of 3450 cm^−1^ [47]. Thus, FT-IR spectra confirmed, at first, the coating of nanoparticles with carbon and, second, its further carboxylation for bio-functionalization of the surface.

The high values of saturation magnetization M_s_ = 79 emu/g in the magnetite NPs and 64 emu/g in the carbon-coated NPs were close to the saturation magnetization of bulk magnetite 84 emu/g (Figure 5). The lower M_s_ value of Fe_3_O_4_@C NPs can be due to a certain fraction of carbon in the mass of NPs, as the total NPs mass is used to determine M_s_ from the measured magnetization. The magnetization curves’ shape with narrow hysteresis loop is typical for single-domain superparamagnetic particles. Each particle has a magnetic moment directed along its magnetic easy axis. In an assemble of noninteracting particles, their easy axes in the absence of magnetic field are oriented randomly, and the total magnetic moments of an assemble equals to zero (point at origin in Figure 5). If we imagine the magnetic moments of particles emitting from one point, then the picture will look like a hedgehog curled up into a ball. When particles are fixed in a space, the application of a magnetic field causes the magnetic moment of each particle to rotate in the direction of the field. To orient the magnetic moment of a particle along the direction of the magnetic field, the energy of the latter must exceed the energy barrier equal to ∆E = KV, where K is the anisotropy energy and V is the particle volume. Additionally, the magnetic field must be higher the larger the angle between the magnetic field and the easy axis directions. In this way, the magnetic moments of the particles will gradually turn towards the direction of the magnetic field until all are settled along the field. This process is described by Langevin’s equation.
(1)$$L\left(x\right)=\coth\left(x\right)-\frac{1}{x}$$
where $x=\frac{g\mu_{B}JB}{k_{B}T}$, *g* is g-factor, $\mu_{B}\;$ is the Bohr magneton, *J* is the total angle moment, *B* =$\;\mu_{0}H$, $\mu_{0}$ is the vacuum magnetic permeability, $k_{B}$ is the Boltzmann’s constant, and *T* is temperature. At *x*→∞, the magnetization saturates and the magnetic moments line up completely in the direction of the applied field. When the magnetic field decreases, the magnetic moments of the particles turn to their easy axes. However, in this case, in a zero field, they will all be oriented in a half-plane, resembling a fan, and thus some remnant magnetization MR of the entire ensemble will be preserved. Namely, such a situation occurs when measuring magnetization of the NPs powder densely packed in a special capsule. Being dispersed in a liquid, NPs participate in free motion—Brownian motion. The magnetic field greatly affects the type of this movement. In a DC gradient field, particles move along the gradient from a smaller field to a larger one and collect at the point of maximum field. This effect is of great interest for solving problems of water purification (and any liquids) from pollution, as it can be used to extract particles with the adsorbed pollutions from the water. In an AC magnetic field, nanoparticles behave like a compass needle, tending to settle in such a way that their magnetic moment is directed in the direction of the magnetic field and, at the same time, begins to rotate. This rotational motion can occur while maintaining a constant orientation of the magnetic moment inside the particle. To implement such a motion, the energy of the interaction of the magnetic field with the magnetization of the particle (Zeeman energy) must exceed the energy of the viscous deceleration of the particle in the fluid. Obviously, the Zeeman energy is greater the greater the magnetic moment of the particle. Such an action can be used in therapy to destroy cancer cells in the body. Rotational motion of free spherical magnetic NPs in a viscous medium under the action of a low-frequency magnetic field was considered, for example, in [35], where functionalized magnetic NPs with a size of 10–50 nm were shown to be capable to generate forces sufficient for cell destruction under an action of low-frequency magnetic fields (1 to 100 Hz) and magnetic induction of hundreds of mT.

### 3.2. Dyes Adsorption by Fe_3_O_4_ NPs

The optical absorption spectra of two anionic CR and EoY and two cationic MB and RhC organic dyes dissolved in distilled water (pH 5.5) in a given concentration were used as the reference spectra to find out the adsorption capacity of the investigated NPs for these dyes. Measurements were carried out at 25 °C. The dye concentration was tied to the absorption intensity at the wavelength corresponding to the maxima in the optical spectra of 495 nm for CR, 514 nm for EoY, 662 nm for MB, and 552 nm for RhC. The Figure 6 shows the optical absorption spectra of the listed dyes. The positions of the maxima are typical for these dye solutions and are close to those determined by other researchers for CR (498 nm) [48], MB (664 nm) [49], EoY (522 nm) [50], and RhC (553 nm) [51]. Thus, the change in dye concentration in the solution could be determined from the change in these lines’ intensities.

In the experiment, a certain concentration of the dye’s aqueous solution was mixed with NPs in an ultrasonic bath for 10 min, after that, NPs with adsorbed dyes were separated from the solution by applying a magnetic field, the intensities of the corresponding lines in the optical absorption spectra of the remaining solution were measured, and the remaining dye’s concentration was determined. 

The values of the adsorption capacity *q_t_* (mg/g) and adsorption percentage (*θ*) of NPs were calculated as follows [52]:(2)$$q_{t}=\frac{(C_{0}-C_{t})V}{m}$$
(3)$$\theta =\frac{\left(C_{0}-C_{t}\right)}{C_{0}}\cdot 100\%,$$
where *C*_0_ and *C_t_* (mg/L) are the initial concentration and concentration of the dye at the contact time *t* (min), *V* (L) is the volume of the solution, and *m* (g) is the mass of the adsorbent (NPs).

The results of the effect of the contact time on the adsorption of Fe_3_O_4_ NPs for the various dyes are shown in Figure 7 at the initial dye concentration of *C*_0_ = 60 mg/L for CR and *C*_0_ = 30 mg/L for the other dyes.

The *q_t_*(*t*) dependencies are different for different dyes: Fe_3_O_4_ NPs adsorb the anionic dye CR very well, although the value of EoY adsorption is two and a half times less than for CR; both dependences are characterized by a short time to reach the equilibrium value. The kinetic of adsorption of the cationic dyes (MB and RhC) is very slow, and saturation is not reached for a very long time.

Usually, two main models were considered to describe kinetics of dye adsorption by NPs [52]—the pseudo-first
(4)$$q_{t}=q_{e}\left(1-e^{-k_{1}t}\right)$$
and pseudo-second order
(5)$$q_{t}=\frac{q_{e}^{2}k_{2}t}{1+q_{e}k_{2}t}$$
where *k*_1_ (1/mg) and *k*_2_ (g/(mg*min) are the rate constants of the sorption reaction in the pseudo-first and pseudo-second orders, respectively, and *q_e_* (mg/g) is the adsorption capacity at equilibrium. In Figure 7a,b, black solid and dotted lines are the results of the pseudo-first-order (PFO) and pseudo-second-order (PSO) nonlinear fitting of adsorption data. The advantages of nonlinear modeling were shown in ref. [53]. Nonlinear modeling was carried out using the OriginPro 2016 program (OriginLab Corporation, Northampton, MA, USA) utilizing the Orthogonal distance regression algorithm. The parameters extracted from the nonlinear fitting of the kinetic data are collected in Table 1. It can be seen that correlation coefficients (*R*^2^) describing the kinetic data for anionic dyes are close for both the pseudo-first and pseudo-second order, but it can be seen from Figure 7 and comparison *q_e_* with the experimental value that it is the pseudo-first-order description that makes it possible to describe the data well, and in this case, the greater thermodynamic benefit of the interaction of the sorbate–sorbent type prevails. In the case of cationic dyes, the kinetic curves are better described by the pseudo-second-order model.

Diffusion equation is often used to describe the adsorption kinetics based on the presentation of kinetic data in coordinates *q_t_* in dependence on *t*^1/2^, which makes it possible to estimate the contribution of internal and external diffusion processes [54]:(6)$$q_{t}=kt^{0.5}+C$$
where *k* (mg/g.min^0.5^) is the intraparticle diffusion rate constant, and *C* (mg/g) is the point of intersection of the linear curve with the *y*-axis; this parameter is related to the thickness of the boundary layer.

From Figure 8, it can be seen that such dependence is described by one straight line only passing through the origin in the case of MB. According to the Weber and Morris model [54], which is intensively used in the analysis of the mechanisms of adsorption of nanoparticles, if such dependence is described by a single straight line passing through the origin, then the adsorption process is controlled only by intra-particle diffusion. A more complex (polylinear) character of this dependence indicates the involvement of intra-particle diffusion processes as well. The parameters of the intra-particle diffusion model are given in Appendix A.

The rapid removal of the dyes in the initial stage, and the nearly horizontal curve at the later stages (Figure 8a), indicate the predominant role of external surface diffusion and the insignificant role of the dye molecules’ diffusion into the particle micropores in the case of anionic dyes.

Concentration dependencies of the adsorption efficiency are shown in Figure 9. It is seen from Figure 9a that the removal efficiency (right axis) of CR is very high at low dye concentration, up to 90% for *C*_0_ = 50 mg/L, and drops gradually up to 60 % with a further increase in the dye concentration. Instantaneous adsorption at low dye concentrations corresponds to the almost vertical initial part of the adsorption isotherm in Figure 9b. The maximum adsorption capacity at 25 °C in the concentration range studied is 58.9 mg/g for CR.

The adsorption isotherms for the several dyes and the experimental data fittings to models of adsorption from liquid phase are shown in Figure 9. It is seen that Langmuir curves (dotted lines in Figure 9b–d) for monomolecular adsorption well describes the isotherm for all investigated dyes in the low concentration range only, and the equilibrium saturation concentration postulated in this model is not reached at all. According to the monomolecular model, with an increase in the proportion of occupied adsorption sites, it is more difficult for the adsorptive molecules to find a vacant site, and the system comes to the equilibrium state. An increase in adsorption after filling the monolayer and the plateau transformation into an increasing curve may be due to polymolecular adsorption or reorientation of molecules, or an appearance of their associates relative to the NP surface. The polymolecular adsorption model was suggested by S. Brunauer, P. H. Emmett, and E. Teller [55]. Several cases were considered in this model, mentioned now as the BET model. The equation for polymolecular adsorption in Case 3 [56], when the number of adsorbing layers n = ∞, is as follows:(7)$$q=q_{m}\frac{K_{S}C_{e}}{\left(1-K_{L}C_{e}\right)\left(1-K_{L}C_{e}+K_{S}C_{e}\right)},$$
which for n = 1 coincides with the Langmuir equation
(8)$$q=q_{m}\frac{K_{S}C_{e}}{1+K_{S}C_{e}}.$$

In these equations, *K_S_* (L/mg) and *K_L_* (L/mg) are equilibrium constants of adsorption for the first and upper layers, correspondingly, *C_e_* (mg/L) is the residual dye concentration, and *q_m_* (mg/g) is the maximal adsorption capacity. For all dyes investigated here, experimental data are well described in the frames of the BET model (Figure 9b–d). Isotherm parameters calculated for CR according to Langmuir and BET models are shown in Table 2. The low value of the correlation coefficient *R*^2^ in the case of the Langmuir modeling and *R*^2^ value close to unity confirm the predominantly polymolecular nature of adsorption in the case under consideration. For other dyes, results are analogous.

Thus, adsorption by Fe_3_O_4_ NP can be used in solutions containing complex pollutants; they show preferential absorption of anionic dyes (especially CR), but they also adsorb cationic dyes, albeit at a lower capacity and rate. The adsorption processes on Fe_3_O_4_ NPs are well described on the bases of the theory of polymolecular (multilayer) adsorption.

### 3.3. Dyes Adsorption by Fe_3_O_4_@C NPs

Magnetite NPs coated with carbon shell showed selective adsorption of cationic dyes: they did not absorb anionic dyes CR and EoY at all, as well as a number of other dyes tested by us (methyl orange and aniline yellow). The magnitude and rate of adsorption of cationic dyes show larger values compared with Fe_3_O_4_ NPs without shell. Figure 10 shows the kinetic curves of the adsorption of MB and RhC. Results of the nonlinear curve fitting of the kinetic adsorption data are collected in Table 3.

It can be seen that the kinetic data for MB are described more closely by the pseudo-first-order model, while the adsorption of RhC is closer to being described by the pseudo-second order.

Interestingly, the adsorption isotherms of the two cationic dyes MB and RhC; are different (Figure 11). In the first case, it is closer to the Friendlich model [57] (also monomolecular as the Langmuir model) and, in the second case, to the BET model of polymolecular adsorption. The proximity to the Freundlich model shows that adsorbate molecules can interact with each other on the surface of the adsorbent, and the increase in adsorption does not stop after the formation of a monomolecular layer, in contrast to the Langmuir model. For the case of RhC, the adsorption isotherm has a polymolecular character (Figure 11b).

### 3.4. Comparison and Mechanism

Now, one can compare the adsorption capacity of the studied NPs with some of theliterature data. The large variation in the adsorption capacity values obtained by different authors (Table 4) is due to not only the variety of materials studied but also to the influence of various factors on the adsorption mechanism and their complex accounting in nanoscale systems. Note that the S-type sorption isotherms obtained here for CR, RhC, and EoY on Fe_3_O_4_ NPs and for RhC on Fe_3_O_4_@C NPs have not been observed by other authors who studied the adsorption of analogous dyes on magnetic nanoparticles. Additionally, in the majority of articles, kinetic data are well described by the pseudo-adsorption equation of the second order, and isotherms are close to the Langmuir model. Only in the work in [58], the authors propose a model of multilayer sorption with calculated thermodynamic parameters close to physical sorption, although the results are described by a second-order pseudo-equation and are in good agreement with the Langmuir model. As concerns the capacity value, both types of the studied NPs demonstrate low capacity for Methylene Blue. For the other three dyes, the absorption capacity is close to the results of other authors which, in combination with high magnetization, allows us to consider them as successful objects for studying promising adsorbents removed from water using a magnetic field.

The FT-IR spectra of Fe_3_O_4_ NPs and Fe_3_O_4_ NPs after adsorption of CR are shown in Figure 12. In the spectrum of nanoparticles after adsorption, changes in line intensities are noticeable, compared with the spectrum of particles before adsorption, and new features appeared. After adsorption of the dye, a line at 1046 cm^−1^ is clearly visible, characteristic of the formation of C=O bonds. In ref. [19], analogical peaks at 878 cm^−1^ and 1046 cm^−1^ appeared in the FT-IR Fe_3_O_4_@C spectra after MB adsorption, which were attributed to C-H in the benzene ring and the rocking vibration -CH_3_. Note that the CR molecule contains six benzene rings in its structure, and similar features are clearly distinguishable on the FT-IR spectrum of our Fe_3_O_4_ NPs after CR adsorption. Besides, slightly visible traces at 1178; at 1228 cm^−^^1^, can be referred to the asymmetry stretching vibration of the S-O (SO_3_-H) group-characteristic of the CR dye [64]. In ref. [60], some shifts of the characteristic peaks and a decrease in their amplitude after the adsorption process were also noted, which the authors attribute to the electrostatic interaction of the adsorbent surface with the RC dye. Thus, the observed changes, i.e., the shift of the peaks and the appearance of new peaks, confirm the successful adsorption of the Congo Red dye on the surface of the Fe_3_O_4_ NPs. Similar results are obtained for other cases.

Differences in the adsorption capacity of the Fe_3_O_4_ and Fe_3_O_4_@C NPs with respect to cationic and anionic dyes indicate a significant difference between the adsorption mechanisms in these two cases. The main mechanisms of adsorption of the cationic dye MB on the carbon shell are considered, for example, in [61]. These are electrostatic interactions, hydrogen bonds, and chemical bonds between dye molecules and adsorbent. The presence of functional groups (such as -OH, COOH, and -CO) on the carbon surface provides a suitable platform for inducing electrostatic attraction between the cationic dyes and the adsorbent. Since, in our case, carbon-coated NPs exhibit selective adsorption of cationic dyes, in the case of Fe_3_O_4_@C NPs, electrostatic interaction predominates. A possible mechanism for the interaction of NPs with MB molecules is schematically shown in Figure 13a.

More complex situation takes place in the case of Fe_3_O_4_ NPs, they adsorb both cationic and anionic dyes, but a strong difference is observed in the NPs’ adsorption capacity with respect to different representatives of the same group. However, the adsorption isotherms have the same character for all studied dyes. The initial steep rise of the isotherm becomes gentler, then their steepness increases again. Qualitatively, it can be assumed that the decrease in steepness is associated with a decrease in the number of free places on the surface (which corresponds to the Langmuir model), and its subsequent increase is due to ever-increasing adsorption in the second and subsequent layers. Thus, it can be assumed that interaction occurs not only between the NPs surface and the arriving molecules but also between molecules arriving from the liquid phase and previously adsorbed molecules. The change in the electron density in arriving molecules is synchronized with the change in the electron density of already adsorbed molecules. Both orientational (in the case of polar molecules) and induced (in the case of charge induction in a molecule approaching the adsorbed layers) interactions arise. The contribution of the longitudinal interactions of molecules is negligibly small compared with perpendicular interactions. Simultaneously, adsorption proceeds on the surface of a solid body and on previously adsorbed molecules of the first, second, third or other layers, and only for the first layer the average residence time of molecules in it is longer than for molecules in other layers, where it is the same for any layer. On the surface, thus, vertical complexes are formed, consisting of one, two, three, etc., molecules and, in addition, there are free places. Such a process is shown schematically in Figure 13b.

### 3.5. Desorption and Reusability Studies

To establish the possibility of reusing the particles as adsorbents, desorption experiments were carried out. The regeneration of the Fe_3_O_4_ NPs after adsorption of all dyes was carried out by washing several times (4–5 times) in ethanol until the solution was transparent. The Fe_3_O_4_@C NPs after adsorption of MB were regenerated first by treatment in a 0.01 mol^−1^ HCl solution (5 min) and subsequent washing in ethanol several times (2–3 times) until the solution was transparent. Then, the NPs washed in deionized water were dried (6 h. at room temperature) and used in subsequent rounds.

The adsorption–desorption cycle was repeated for times using the same NPs, and the results for CR adsorption by Fe_3_O_4_ NPs and MB adsorption by Fe_3_O_4_@C NPs are shown in Figure 14. It was observed that the adsorption capacities in both cases decreases by no more than 10% after four cycles. The percentage of CR removal was 85.6, 82.5, 79.98, and 78, and for MB, 65.3, 64.9, 64.2, and 58.3% from cycle to cycle. The NPs thus showed a high regenerative capacity after four cycles of adsorption–desorption. The easy desorption is most likely due to the predominance of physical adsorption processes.

### 3.6. Magneto-Mechanical Destruction of Ehrlich Ascites Carcinoma Cells by Fe_3_O_4_@C NPs

For experiments with living cells, aptamer-functionalized NPs were incubated for 30 min with Ehrlich ascitic carcinoma cells and then exposed to a low-frequency alternating magnetic field. Destroyed cells are clearly visible in the microscope chamber as blue spots (Figure 15a).

To reveal the combined effect of immobilization of magnetic NPs by aptamers and a low-frequency magnetic field on the death of model cancer cells, we performed a statistical analysis of four cases: (1) control cells; (2) cells mixed with NP functionalized by aptamers without applying a magnetic field; (3) cells mixed with NP functionalized by aptamers after exposure to an alternating magnetic field; and (4) a mixture of cells with initial Fe_3_O_4_@C NPs after action of an alternating magnetic field. The percentage of disrupted ascites cells in the test samples was statistically evaluated on five independent probes of each sample, processing at least 10 different images of each so that the average number of particles in one image was 100–180 (Figure 15b).

Experimental data showed that the use of magnetic NPs increased the amount of disrupted cells. More than 28% of cells were dead in cells with aptamer-functionalized Fe_3_O_4_@C NPs after exposure to the alternating magnetic field compared with the ~6% control cell death. The percentage of destroyed cells in all cases is greater than in the control cells, which indicates the attachment of nanoparticles to cells in all cases. In our experiment, the frequency of the change in the magnetic field was determined by the frequency of the electric alternating current in the network and was 50 Hz. Experiments carried out by other authors, for example, in [65], showed that the most effective cell destruction frequency range is 10–40 Hz. However, the results of this work, when using a field of 90 Oe and a frequency of 50 Hz, are comparable with ours (28%), so the cell death of glioblastoma multiforme (aggressive form of brain cancer) was 8 and 23% when using magnetic particles coated with IrG and IL13, respectively. Based on the results presented in [65], and in our case, we can assume an increase in cell mortality of 80–90% with a decrease in the frequency of the applied magnetic field.

## 4. Conclusions

Thermal decomposition was used to synthesize Fe_3_O_4_ and Fe_3_O_4_@C nanoparticles (NPs) uniform in shape and size with an average size of 15 ± 2 nm, and their morphology, structure, and magnetic properties were studied. The formation of a carbon shell around the magnetite core and the modification of the NPs surface by carboxylated groups were confirmed by IR-Fourier spectra. The obtained high value of the saturation magnetization (more than 63 emu/g) of nanoparticles is their important advantage, since it facilitates the extraction of particles after the adsorption of pollutants from the treated water.

The NPs capability to adsorb the organic dyes Methylene Blue (MB), Rhodomine C (RhC), Congo Red (CR), and Eosin Y (EoY) was studied. It is shown that the adsorption kinetics depends not only on the type of dye but also on its composition. The highest capacity (58 mg/g) was observed upon adsorption of CR on Fe_3_O_4_ NPs. At dye concentrations up to 50 mg/L, more than 90 percent of the dye was removed instantly. At higher dye concentrations, the sorption reaction reached equilibrium in approximately 75 min. Nonlinear modeling showed that, in the case of Fe_3_O_4_ NPs, the kinetic data of the sorption reaction are better described by a pseudo-first-order equation for anionic dyes and a pseudo-second-order equation for cationic dyes. Adsorption isotherms in the region of low pollutant concentrations fit well into the Langmuir model for all the dyes studied, i.e., sorption can be due to a monomolecular model. With an increase in the concentration of the dye in the solution, the trend changes, and the experimental value of *q_e_* for all the dyes used continues to grow, that is, the adsorption process is more adequately described by the BET equation for the polymolecular adsorption model.

Fe_3_O_4_@C NPs do not adsorb anionic dyes, and thus have a selective adsorption capacity for cationic dyes. For MB, the kinetic data are more accurately described by a pseudo-first-order equation and, for RhC, by a pseudo-second-order equation. The MB adsorption isotherm is well described by a monomolecular process, and it is better in the Freundlich model than in the Langmuir model.

The results obtained for bio-functionalized Fe_3_O_4_@C particles can be used for targeted magneto-mechanical destruction of cancer cells. So, the percent of disrupted cells as results of interaction with Fe_3_O_4_@C NPs functionalized with aptamers after exposure to a low-frequency magnetic field was 27%, compared with the number of dead control cells of 6%. Thus, the investigated nanoparticles seem to be perspective for their use in water treatment technologies on one hand and interesting nano-objects for the fundamental study of the dye adsorption processes on the other hand. The first experiments on the use of these particles for the targeted magneto-mechanical destruction of cancer cells also gave encouraging results.

## Figures and Tables

**Figure 1 materials-16-00023-f001:**
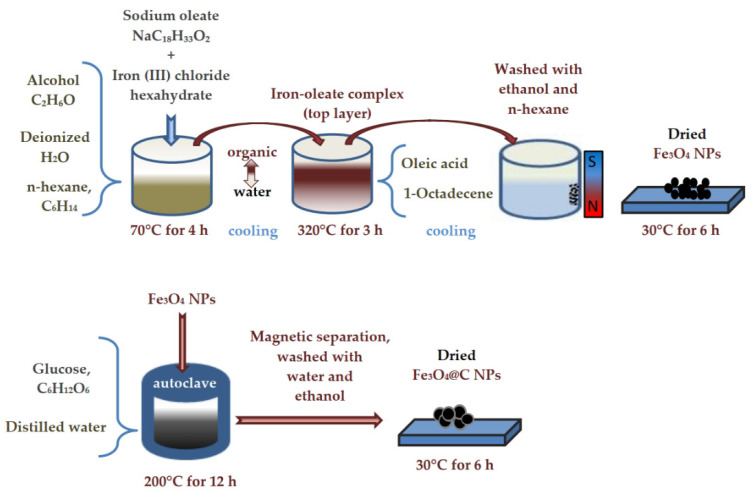
The scheme of NPs’ synthesis.

**Figure 2 materials-16-00023-f002:**
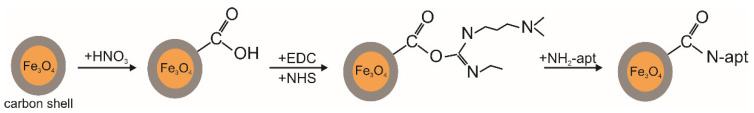
The scheme of the functionalization of carbon shell by aptamers. EDS and NHS—buffer solution with Ca^2+^ and Mg^2+^, pH = 7.4, containing: 1-ethyl-3-(3-dimethylaminopropyl) carbodiimide (1 mg) N-hydroxysuccinimide (1 mg) for 30 min.

**Figure 3 materials-16-00023-f003:**
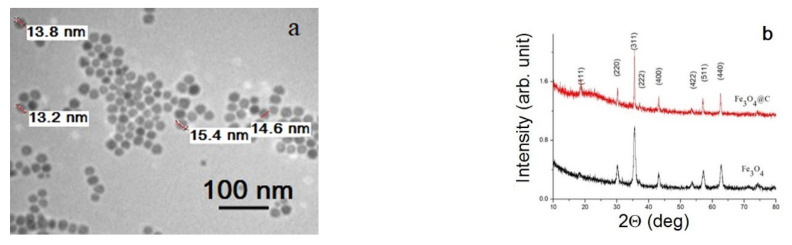
TEM image of the initial Fe_3_O_4_ NPs (**a**) and XRD patterns for Fe_3_O_4_ and Fe_3_O_4_@C NPs (**b**).

**Figure 4 materials-16-00023-f004:**
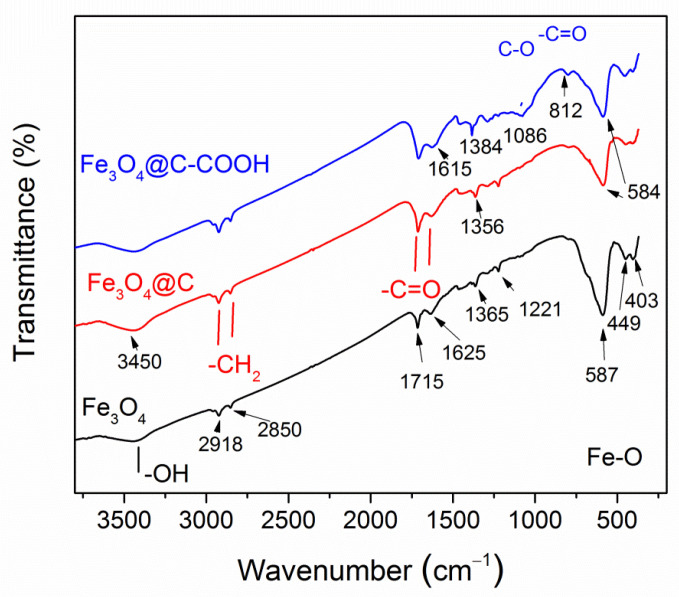
FT-IR spectra of synthesized Fe_3_O_4_, Fe_3_O_4_@C NPs, and Fe_3_O_4_@C NPs after nitric acid treatment.

**Figure 5 materials-16-00023-f005:**
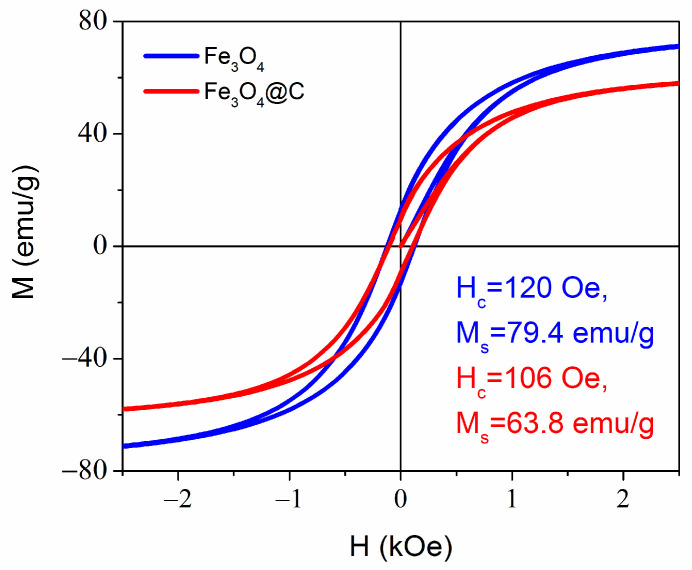
Room temperature magnetization curves for Fe_3_O_4_ and Fe_3_O_4_@C NPs. The remnant magnetization M_r_ = 12.7 emu/g and 9.7 emu/g for Fe_3_O_4_ and Fe_3_O_4_@C NPs, respectively.

**Figure 6 materials-16-00023-f006:**
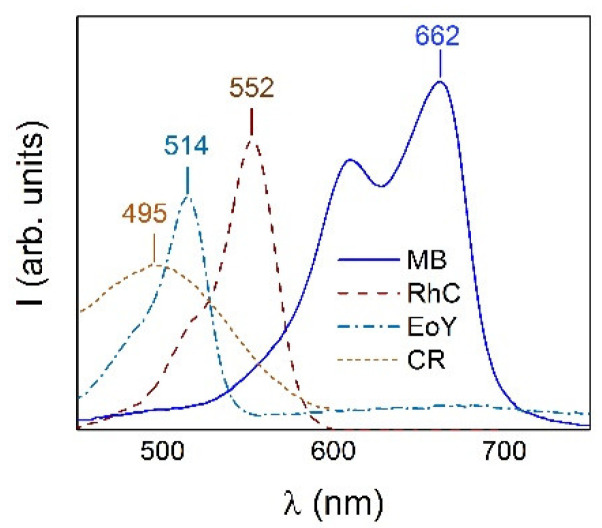
Optical absorption spectra of dye solutions (CR, EoY, RhC, and MB) with indication of the wavelength of the maxima that were used as reference to determine the concentration of dyes in solutions.

**Figure 7 materials-16-00023-f007:**
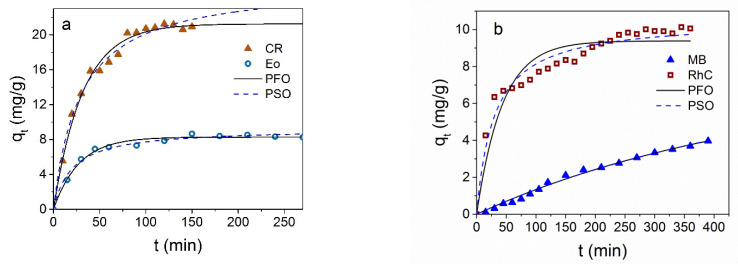
The *q_t_* dependence on the Fe_3_O_4_ contact time with CR and EoY (**a**), MB, and RhC (**b**). Black solid lines are nonlinear fitting of the results to the pseudo-first-order (PFO) model; dotted lines are the results nonlinear fitting to the pseudo-second-order (PSO) model. Experimental conditions: 25 °C, *C*_0_ = 60 mg/L for CR and *C*_0_ = 30 mg/L for other dyes, *m* (NPs) ≈ 3 mg in *V* = 1.5 mL.

**Figure 8 materials-16-00023-f008:**
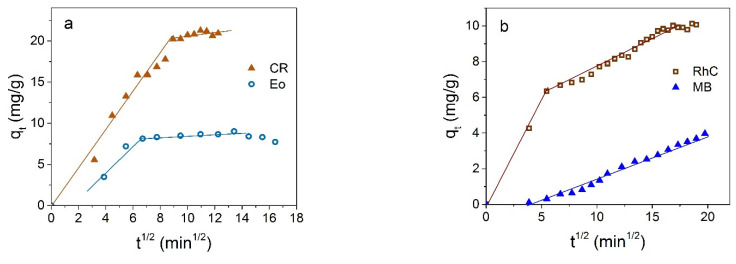
Representation of kinetic adsorption data of Fe_3_O_4_ NPs in coordinates *q_t_* vs. *t*^1/2^ for anionic (**a**) and cationic (**b**) dyes.

**Figure 9 materials-16-00023-f009:**
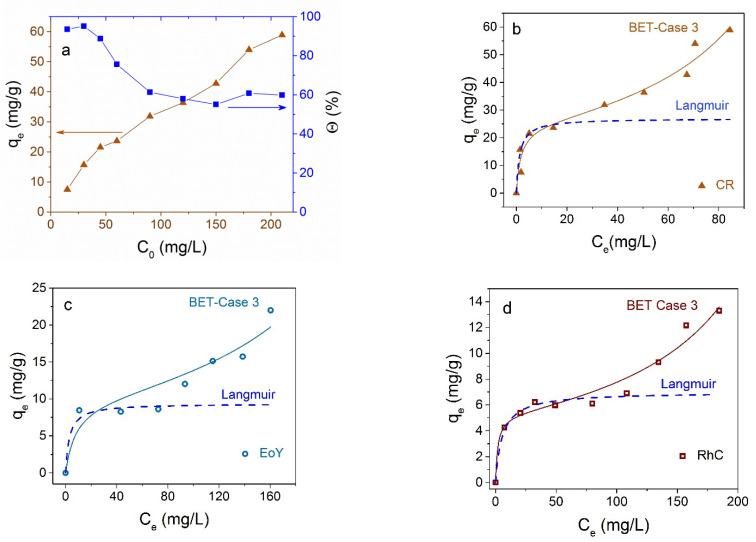
Adsorption isotherms of CR on Fe_3_O_4_ NPs at 25 °C in depending on initial *C*_0_ (**a**) and residual *C_e_* (**b**) dye concentration. The right axis in (**a**) shows the percentage of dye removal. Experimental adsorption isotherms of EoY (**c**) and RhC (**d**) are shown by circles and squares, correspondingly. Adsorption isotherms according to the BET and Langmuir models are shown in panels (**b**–**d**) by solid and dashed curves, correspondingly.

**Figure 10 materials-16-00023-f010:**
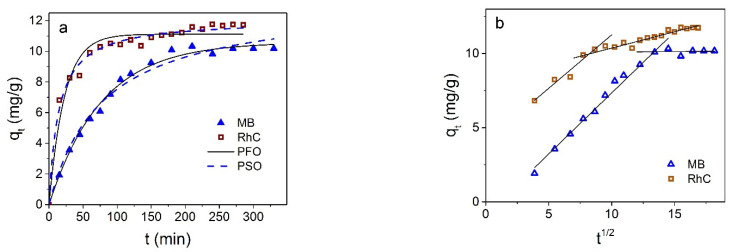
(**a**) *q_t_* dependence on the Fe_3_O_4_@C contact time with MB and RhC; solid line is fitting to the pseudo-first-order (PFO) and dotted line to pseudo-second-order (PSO) models. (**b**) Presentation of kinetic data in coordinates *q_t_* from *t*^1/2^.

**Figure 11 materials-16-00023-f011:**
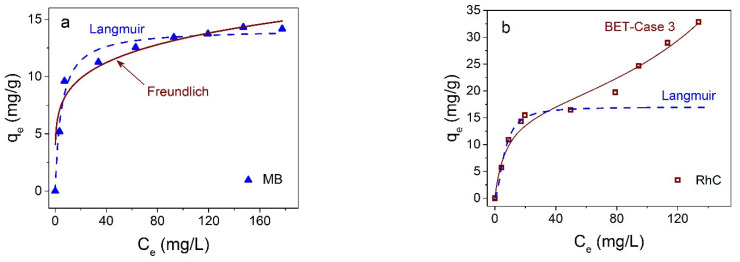
Adsorption isotherms of MB and RhC on Fe_3_O_4_@C NPs at at 25 °C—(**a**,**b**), correspondingly. Solid lines correspond to fitting to Freundlich (**a**) and BET (**b**) models, dotted lines to Langmuir model.

**Figure 12 materials-16-00023-f012:**
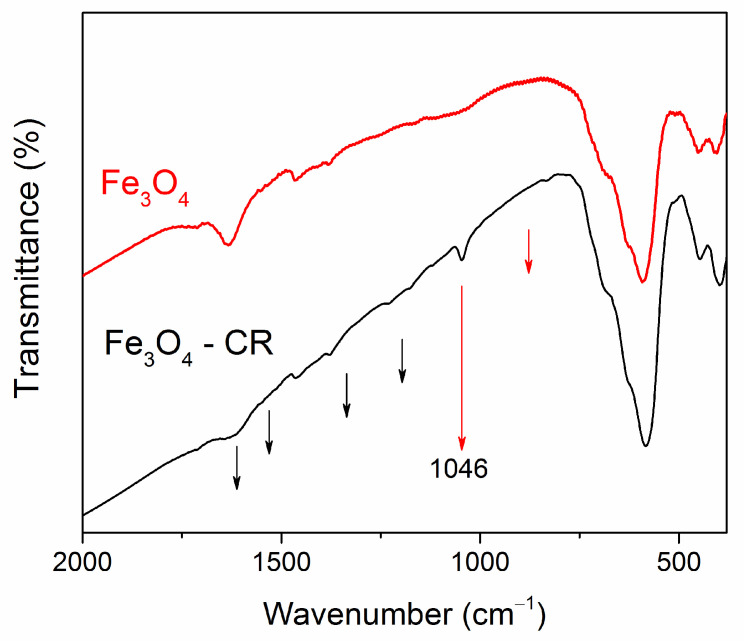
FT-IR spectra of Fe_3_O_4_ NPs before and after adsorption of CR. The black arrows indicate the position of the CR features, which correspond to data of Ref. [64]. The red lines indicate the positions of new lines in the spectrum of the Fe_3_O_4_@C nanoparticle after dye adsorption, correlating with data of Ref. [19].

**Figure 13 materials-16-00023-f013:**
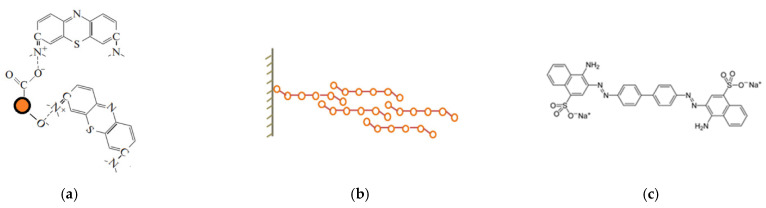
(**a**) Possible mechanism of electrostatic interaction of the MB molecule with the carbon shell, (**b**) mechanism of physical adsorption, and (**c**) molecule of the CR.

**Figure 14 materials-16-00023-f014:**
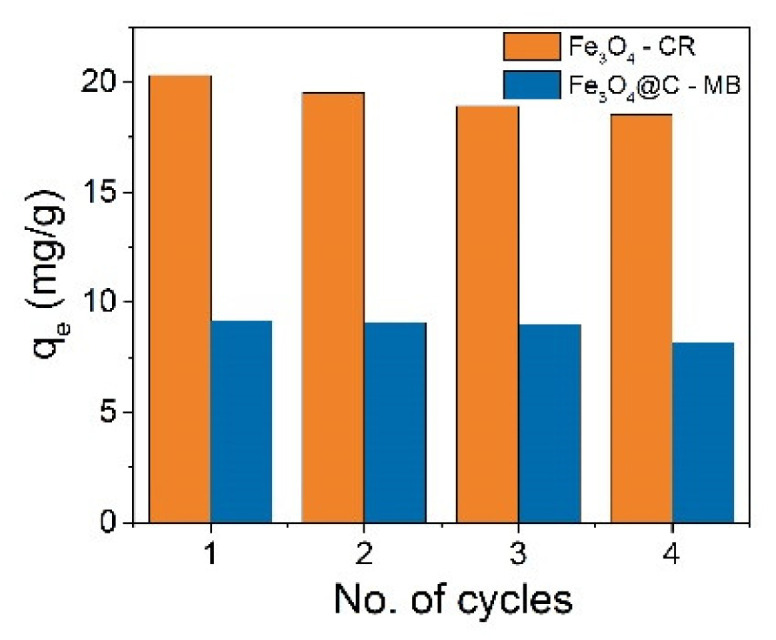
Reusability of Fe_3_O_4_ and Fe_3_O_4_@C NPs for adsorption Congo Red (CR) and Methylene Blue (MB), correspondingly.

**Figure 15 materials-16-00023-f015:**
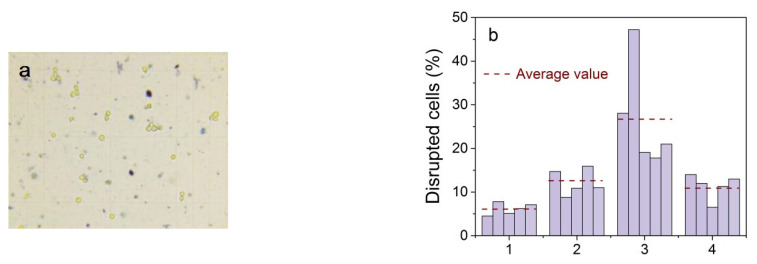
A typical picture in a microscope chamber; yellow objects with a blue border are living cells, blue spots are destroyed cells (**a**). Data of statistical processing of living and destroyed cells on five independent probes of each sample (**b**). (1) control cells (natural mortality), (2) a mixture of cells and Fe_3_O_4_@C NPs functionalized with aptamers without magnetic field exposure; (3) a mixture of cells and Fe_3_O_4_@C NPs with functionalized aptamers after action of a low-frequency magnetic field; and (4) a mixture of cells with initial NPs after action of a low-frequency magnetic field.

**Table 1 materials-16-00023-t001:** Parameters of the best nonlinear fit to the pseudo-first- and pseudo-second-order kinetic models of adsorption data for Fe_3_O_4_ NPs, for experimental conditions of 25 °C, *C*_0_ = 60 mg/L for CR and *C*_0_ = 30 mg/L for other dyes, and *m* (NPs) ≈ 3 mg in *V* = 1.5 mL.

	Parameters of the Nonlinear Fit				Experiment
Dyes	Kinetic Model	*q_e_* (mg/g)	*k*_1_ (1/min) *k*_2_ (g/(mg*min)	*R* ^2^	*q_e_* (mg/g)
CR	PFO	21.249 ± 0.319	0.031 ± 0.0018	0.989	21.6
	PSO	25.960 ± 0.70	0.00133 ± 0.00016	0.988	
EoY	PFO	8.28 ± 0.121	0.036 ± 0.0025	0.987	8.3
	PSO	9.265 ± 0.224	0.0054 ± 0.0008	0.984	
MB	PFO	7.2 ± 0.85	0.002 ± 0.0003	0.992	4.5 *
	PSO	12.58 ± 1.83	0.00009 ± 0.000032	0.992	
RhC	PFO	9.38 ± 0.206	0.023 ± 0.0027	0.889	10.2
	PSO	10.53 ± 0.222	0.003 ± 0.0004	0.958	

* The experimental value of *q_e_* for MB is significantly less than the value given by fitting to pseudo-models due to the fact that for the concentration and time of the experiment used, we did not reach the equilibrium value (see Figure 7b (data for MB)).

**Table 2 materials-16-00023-t002:** Calculated for CR parameters of isotherm for different adsorption models.

CR Dye	BET-Case 3	Langmuir
*K_L_* (L/mg)	0.007 ± 0.0005	–
*K_S_* (L/mg)	0.831 ± 0.0	0.7608 ± 1.136
*q_m_* (mg/g)	23.73 ± 1.56	27.49 ± 20.76
*R* ^2^	0.9695	0.3188

**Table 3 materials-16-00023-t003:** Parameters of the best nonlinear fitting of the experimental adsorption data for Fe_3_O_4_@C NPs to the pseudo-first- and pseudo-second-order kinetic models.

		Parameters of the Nonlinear Fit			Experiment
Dyes	Kinetic Model	*q_e_* (mg/g)	*k*_1_ (1/min) *k*_2_ (g/(mg*min)	*R* ^2^	*q_e_* (mg/g)
MB	PFO	10.59 ± 0.159	0.013 ± 0.0006	0.993	11
	PSO	13.33 ± 0.487	0.0009 ± 0.00014	0.984	
RhC	PFO	11.12 ± 0.166	0.044 ± 0.0045	0.950	12
	PSO	12.09 ± 0.137	0.0059 ± 0.0005	0.987	

**Table 4 materials-16-00023-t004:** Comparison of several dyes’ adsorption capacities reported in the literature and obtained in the present work.

Adsorbent	Dyes	q_e_ [mg/g]	Refs.
Fe_3_O_4_ NPs	Congo Red	96.46	[59]
Fly ash		22.12	[60]
Fe_3_O_4_ NPs		58	Present work
Fe_3_O_4_@C NPs (30 nm with 2 nm carbon shell)	Methylene Blue	18.52	[19]
Fe_3_O_4_@C magnetic materials (particle size, 1~100 μm)		270.51	[26]
Fe_3_O_4_@C-dots		124.9	[61]
B-Fe_3_O_4_@C (~3 μm)		42.11	[21]
Fe_3_O_4_@C NPs		15	Present work
NiZnAl-LDH, ZnAl-LDH, NiAl-LDH	RhB	52–97	[4]
Nano iron oxide–modified biochar		286.4	[62]
Fe_3_O_4_ NPs	RhC	14	Present work
Fe_3_O_4_@C NPs		35	Present work
Nanoplates γ-Al_2_O_3_	EoY	6	[63]
Fe_3_O_4_ NPs		22	Present work

## Data Availability

The data presented in this study are available on request from the corresponding author. The data are not publicly available since they are a part of ongoing research.

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
