# Peer review of "Core–Shell Fe3O4@C Nanoparticles for the Organic Dye Adsorption and Targeted Magneto-Mechanical Destruction of Ehrlich Ascites Carcinoma Cells"

_materials, 2022, doi:10.3390/ma16010023_

Round 1
Reviewer 1 Report
Manuscript No: materials-2053866
Title: Core-shell Fe3O4@C nanoparticles for the organic dye adsorp-2 tion and targeted magneto-mechanical destruction of Ehrlich 3 ascites carcinoma cells.
Comments
1. In abstract add quantitative data about characterizations and application
2. Ist sentence of introduction need to be revise. Write short sentences for better understanding. It is better to discuss applications of NPs separately one by one.
3. Line 39-45 add appropriate references.
4. In introduction mentioned advantages of adsorption technique for the removal of dyes over others techniques like photo degradation, advances oxidation processes. Authors should refer and cite these articles to fill the gap, Inorganic Chemistry Communications 145 (2022) 110008, Surfaces and Interfaces 34 (2022) 102324.
5. Characterizations of synthesized NPs are not enough, N2 adsorption and XPS should be implemented. Additionally ZPC (zero point charge ) of NPs should be calculated.
6. For λmax cite references for example for EoY cite Korean J. Chem. Eng., 39(1), 216-226 (2022)
7. Cite references for Eq 2-5.
8. Thermodynamic parameters must be considered to predict the spontaneity.
9. Mechanism of adsorption should be discussed along with graphical representation.
10. The results of adsorption capacity of synthesized adsorbent for the removal of anionic and cationic dyes should be compared with latest reported materials
11. FTIR analysis after adsorption should be performed
12. Reusability test for adsorbent should be carried out to check its usability at commercial scale
13. Whole manuscript should be carefully checked for typo, formatting, grammatical and syntax errors before resubmission.
Author Response
Response to Reviewer 1
Comment 1
In abstract add quantitative data about characterizations and application
Answer: We have included quantitative data in the abstract.
Comment 2
I-st sentence of introduction need to be revise. Write short sentences for better understanding. It is better to discuss applications of NPs separately one by one.
Comment 3
Line 39-45 add appropriate references.
Comment 4
In introduction mentioned advantages of adsorption technique for the removal of dyes over others techniques like photo degradation, advances oxidation processes. Authors should refer and cite these articles to fill the gap, Inorganic Chemistry Communications 145 (2022) 110008, Surfaces and Interfaces 34 (2022) 102324.
Answers to comments 2-4
We completely reconsidered the introduction, taking into account the comments of all reviewers. We have included a description of other methods for the water purification, removed unnecessary places that were not related to the essence of the work, added References, including those recommended by reviewers.
Comment 5
Characterizations of synthesized NPs are not enough, N2 adsorption and XPS should be implemented. Additionally ZPC (zero point charge) of NPs should be calculated.
Answer:
Unfortunately, the extremely limited time did not give us the opportunity to carry out these important experiments. Since the results obtained seem us to be rather interesting, we plan to continue this work and carry out these experiments by all means.
Comment 6
For λmax cite references for example for EoY cite Korean J. Chem. Eng., 39(1), 216-226 (2022)
Answer: It is done.
Comment 7
Cite references for Eq 2-5.
Answer: It is done.
Comment 8
Thermodynamic parameters must be considered to predict the spontaneity.
Answer: As in the answer to comment 5, we don't have enough time to do that right now.
Comment 9
Mechanism of adsorption should be discussed along with graphical representation.
Answer: It is done.
Comment 10
The results of adsorption capacity of synthesized adsorbent for the removal of anionic and cationic dyes should be compared with latest reported materials
Answer: It is done.
Comment 11
FTIR analysis after adsorption should be performed
Answer: It is done.
Comment 12
Reusability test for adsorbent should be carried out to check its usability at commercial scale
Answer: Reusability test was carried out and results of these experiments are added to the text.
Comment 13
Whole manuscript should be carefully checked for typo, formatting, grammatical and syntax errors before resubmission.
Answer: We did our best to make it
The authors are grateful to the reviewer for attention to the manuscript
and constructive comments, which helped to improve the work.

Reviewer 2 Report
This manuscript cannot be published in this journal at the present form and major revisions are necessary.
1- The introduction is inappropriate with the results presented in the work?? I suggest changing the introduction and specifying the interest of the prepared material in the field of wastewater treatment.
2- The novelty and practical applicability of this study should be clearly highlighted in the manuscript.
3- SEM analysis would be fruitful tool describing the difference in surface morphology before and after adsorption process. The authors would be appreciated discussing and including this effect in their article.
4- add the units of each parameter of all equations (4), (5), (6) and (7)
5- Add the units of each parameter in the table 1.
6- page 8: Add the value of the experimental adsorbed quantity qe in table 1 and compared it with those calculated by the two models to deduce the most suitable kinetic model.
7- The authors are invited to confirm the adsorption of the dyes on the surface of the adsorbent by an FTIR analysis of the adsorbent after adsorption
8- Add the intra-particle diffusion model equation and their parameters
9- what is the used mass of the adsorbent in the adsorption tests?
10- What about the reusability of synthesized adsorbent.
11- Conclusion: This section is too long and should be systematized.
12- Please read these references
Journal of Molecular Liquids 335 (2021) 116560
Separation Science and Technologie (Philadelphia), 2022, 57(4), pp. 542–554
Author Response
Response to Reviewer 2
Remark 1. The introduction is inappropriate with the results presented in the work?? I suggest changing the introduction and specifying the interest of the prepared material in the field of wastewater treatment.
Answer: We completely reconsidered the introduction, taking into account the comments of all reviewers. We have included a description of other methods for the water purification, removed unnecessary places that were not related to the essence of the work, added References, including those recommended by reviewers.
Remark 2. The novelty and practical applicability of this study should be clearly highlighted in the manuscript.
Answer: It is done.
Remark 3. SEM analysis would be fruitful tool describing the difference in surface morphology before and after adsorption process. The authors would be appreciated discussing and including this effect in their article.
Answer: Unfortunately, the extremely limited time did not give us the opportunity to carry out this important experiment. Since the results obtained seem us to be rather interesting, we plan to continue this work and carry out this experiment by all means.
Remark 4. - add the units of each parameter of all equations (4), (5), (6) and (7)
Answer: we have added units of each parameter
Remark 5. Add the units of each parameter in the table 1.
Answer: we have added units of each parameter in the Table 1.
Remark 6. page 8: Add the value of the experimental adsorbed quantity qe in table 1 and compared it with those calculated by the two models to deduce the most suitable kinetic model.
Answer: we added the experimental value of qe to the table 1 and 3, made a comparison, and indicated the most appropriate model in the text.
Remark 7. The authors are invited to confirm the adsorption of the dyes on the surface of the adsorbent by an FTIR analysis of the adsorbent after adsorption
Answer: It is done
Remark 8. Add the intra-particle diffusion model equation and their parameters
Answer: We have added the intra-particle diffusion model equation to the text and listed the parameters to Table S1.
Remark 9. what is the used mass of the adsorbent in the adsorption tests?
Answer: We used 3 mg of NPs per 1.5 ml of dye solution. It is written in the text.
Remark 10. What about the reusability of synthesized adsorbent.
Answer: The results of the desorption experiment are inserted in the text of the article.
Remark 11. Conclusion: This section is too long and should be systematized.
Answer: We have shortened this section.
Remark 12. Please read these references Journal of Molecular Liquids 335 (2021) 116560
Separation Science and Technologie (Philadelphia), 2022, 57(4), pp. 542–554
Answer: Thank you, we got acquainted with the articles, and took them into account in the discussion.
The authors are grateful to the reviewer for attention to the manuscript
and constructive comments, which helped to improve the work.

Reviewer 3 Report
In the manuscript titled “Core-shell Fe3O4@C nanoparticles for the organic dye adsorption and targeted magneto-mechanical destruction of Ehrlich ascites carcinoma cells” Fe3O4 nanoparticles were prepared via thermal decomposition, the structural, morphological and magnetic properties are studied. The obtained materials are used for adsorption of organic dyes and the adsorption process are described.
The paper is sound and can be accepted for publication in the journal after minor revision:
1) The novelty of this work is missing in the abstract
2) The synthesis method of Fe3O4 is too long and must shortened. It will be better if the authors propose a descriptive scheme for the preparation method.
3) Please numbered the subtitle of Synthesis of materials as 2.1, do the same for the other subtitle.
4) For the dyes adsorption, authors could add a full comparison between the results obtained in this work with other works published elsewhere.
5) The conclusion is too long and should be shortened
Author Response
Response to Reviewer 3
Thank you for your attention to our manuscript and constructive comments. We have finalized the article, and responded to all your comments. The comments of other reviewers were also taken into account.
- The novelty of this work is missing in the abstract
Answer: It is done.
- The synthesis method of Fe3O4 is too long and must shortened. It will be better if the authors propose a descriptive scheme for the preparation method.
Answer: It is done.
- Please numbered the subtitle of Synthesis of materials as 2.1, do the same for the other subtitle.
Answer: It is done.
- For the dyes adsorption, authors could add a full comparison between the results obtained in this work with other works published elsewhere.
Answer: It is done.
- The conclusion is too long and should be shortened
Answer: It is done.
Reviewer 4 Report
In my opinion, the submitted manuscript is suitable for publication after correction, The manuscript is correctly written, but it can be improved.
Remarks:
- more information about sorbents can be added in the introduction, including one of the possibilities of using microorganisms as sorbent components that can be used as a sorbent for removing dyes. Here are examples of using biocomposite as a sorbent doi.org/10.3390/ma14237482 and doi.org/10.1016/j.jclepro.2019.117624 There are microorganisms capable of degrading various dyes.
- I suggest changing the keyword part because they repeat with words in the title.
- The description of the FTIR analysis needs to be improved. There is no full discussion of the results. Describe, for example, emerging shifts in peaks.
- The manuscript should be reviewed again and corrected, e.g. Line 498 - no dot at the end of the sentence.
- The discussion of the results should be expanded. No discussion of results with other studies (especially equilibrium and kinetic studies).
Author Response
Thank you for your attention to our manuscript and constructive comments. We have finalized the article, and responded to all your comments. The comments of other reviewers were also taken into account.
Remarks:
- more information about sorbents can be added in the introduction, including one of the possibilities of using microorganisms as sorbent components that can be used as a sorbent for removing dyes.
Here are examples of using biocomposite as a sorbent doi.org/10.3390/ma14237482 and
doi.org/10.1016/j.jclepro.2019.117624 There are microorganisms capable of degrading various dyes.
Answer: It is done.
- I suggest changing the keyword part because they repeat with words in the title.
Answer: It is done.
- The description of the FTIR analysis needs to be improved. There is no full discussion of the results.
Describe, for example, emerging shifts in peaks.
Answer: It is done.
- The manuscript should be reviewed again and corrected, e.g. Line 498 - no dot at the end of the
sentence.
Answer: It is done.
- The discussion of the results should be expanded. No discussion of results with other studies
(especially equilibrium and kinetic studies).
Answer: It is done.

Round 2
Reviewer 1 Report
Accepted in present form
Reviewer 2 Report
The revised version can be accepted for publication